# Fundamental Relation for Gas of Interacting Particles in a Heat Flow

**DOI:** 10.3390/e25091295

**Published:** 2023-09-04

**Authors:** Robert Hołyst, Karol Makuch, Konrad Giżyński, Anna Maciołek, Paweł J. Żuk

**Affiliations:** 1Institute of Physical Chemistry, Polish Academy of Sciences, Kasprzaka 44/52, 01-224 Warszawa, Poland; kgizynski@ichf.edu.pl (K.G.); amaciolek@ichf.edu.pl (A.M.); pzuk@ichf.edu.pl (P.J.Ż.); 2Max-Planck-Institut für Intelligente Systeme Stuttgart, Heisenbergstr. 3, D-70569 Stuttgart, Germany; 3Department of Physics, Lancaster University, Lancaster LA1 4YB, UK

**Keywords:** thermodynamics, entropy, steady-state, excess heat, nonequilibria

## Abstract

There is a long-standing question of whether it is possible to extend the formalism of equilibrium thermodynamics to the case of nonequilibrium systems in steady-states. We have made such an extension for an ideal gas in a heat flow. Here, we investigated whether such a description exists for the system with interactions: the van der Waals gas in a heat flow. We introduced a steady-state fundamental relation and the parameters of state, each associated with a single way of changing energy. The first law of nonequilibrium thermodynamics follows from these parameters. The internal energy *U* for the nonequilibrium states has the same form as in equilibrium thermodynamics. For the van der Waals gas, $U(S^{*},V,N,a^{*},b^{*})$ is a function of only five parameters of state (irrespective of the number of parameters characterizing the boundary conditions): the effective entropy $S^{*}$, volume *V*, number of particles *N*, and rescaled van der Waals parameters $a^{*}$, $b^{*}$. The state parameters, $a^{*}$, $b^{*}$, together with $S^{*}$, determine the net heat exchange with the environment. The net heat differential does not have an integrating factor. As in equilibrium thermodynamics, the steady-state fundamental equation also leads to the thermodynamic Maxwell relations for measurable steady-state properties.

## 1. Introduction

The determination of energy and its changes induced by heat or work are necessary to understand systems such as combustion engines or the Earth’s atmosphere with weather phenomena. When an equilibrium state approximates a system state, thermodynamics allows one to predict the system’s behavior by using energy as a function of a few parameters of state and a few principles. In particular, the first law of thermodynamics [1] represents a global energy conservation law. The energy, U(S,V,N), is a function of entropy, *S*, volume, *V*, and the number of molecules, *N*. Each variable is related to one independent way of energy exchange: heat, work, and change in the amount of matter.

In equilibrium thermodynamics, U(S,V,N) is called a fundamental relation [1]. Müller, in his book on the history of thermodynamics, explains its role as follows. “Its importance can hardly be overestimated; it saves time and money and it is literally worth billions to the chemical industry, because it reduces drastically the number of measurements” [2]. In the same spirit, Daivis claims that “For this reason alone, it would seem very desirable to establish a formalism for the thermodynamics of nonequilibrium steady-states that has the same attractive features” [3].

However, a similarly simple theory does not exist for nonequilibrium systems in steady-(stationary-)states. There is no description similar to thermodynamics that grasps the energy transfer to the system in terms of a few global parameters. One of the most-straightforward nonequilibrium cases is a steady heat flow. The appearance of the heat flow opens many research directions belonging to various fields of physics. Rational and extended thermodynamics focus on local transport equations [4]. Irreversible thermodynamics formulates thermo-hydrodynamic descriptions with local equations of state and mass, momentum, and energy balance [5]. Sometimes, it is possible to represent governing equations in terms of variational principles [6,7,8,9], which determine the profile of thermodynamic fields (such as temperature).

The issue closely related to the studies mentioned above is whether we can represent the energy of the nonequilibrium system as a function of a few global parameters. The answer to this question would lead to a description similar to classical equilibrium thermodynamics. The existence of such a thermodynamic-like description for steady-state systems has been considered in various studies [3,7,10,11,12,13]. The progress [14,15,16,17] in this field is limited to small temperature differences and low heat fluxes. The recent papers on this topic carry the conviction that general rules exist in nonequilibrium thermodynamics. However, skepticism regarding the usefulness of the equilibrium-based entropy [18] or even the existence of a description in terms of thermodynamic-like potentials [19] also appears.

Lieb and Yngwasson [18] expressed skepticism regarding the use of entropy by suggesting heat as a primary quantity. It requires a generalization of heat for steady-states. However, how can it be generalized, e.g., for a steady gas between two plates with heat flow in a perpendicular direction? Thermo-hydrodynamic equations describe the system, so the heat flowing through the surface is well-defined. This applies both for a steady-state and when the system passes from one stationary state to another. In a steady-state, the same amount of heat enters through one plate and leaves on the opposite side. The net heat vanishes. However, the net heat may flow to the system during the transition between steady-states. This reasoning leads to a concept of heat measured in transition between stationary (steady-)states. It is a particular case of the excess heat discussed by Oono and Paniconi [20]. In 2019, Nakagawa and Sasa [21] noticed that the excess heat concept defined by Oono and Paniconi had yet to be further utilized by other researchers. We adopted the term net (or excess) heat to name the heat that enters the system and changes its internal energy during the transition between steady-states. We note that, in the literature, the excess heat has other meanings [22].

Our recent investigations of an ideal gas in a steady-state with a heat flow showed a surprising result [23]. We proved that the net heat has an integrating factor and rigorously calculated nonequilibrium “entropy” and nonequilibrium temperature. This entropy determines steady adiabatic insulation during transitions between stationary states. However, it is not clear whether the nonequilibrium entropy exists beyond the ideal gas approximation. We continued the research to formulate global steady thermodynamics using van der Waals gas as an example of an interacting system. First, from the thermo-hydrodynamic equations, we derived the global energy balance. Next, we show that it is possible to represent the non-homogeneous van der Waals gas in a heat flow with equations formally identical to the equations of state for the van der Waals gas in equilibrium. This procedure (named mapping) defines the parameters of the state for the nonequilibrium system in the steady-state. We also show that the net heat does not have an integrating factor as proposed by Oono and Paniconi [20]. Instead, the net heat is represented by two independent thermodynamic parameters of state in the van der Waals gas. Moreover, we discuss the experimental determination of the introduced nonequilibrium parameters of state and experimental predictions based on the steady-state fundamental relation.

## 2. Van der Waals Gas in Equilibrium

We considered the van der Waals fluid described by the following fundamental thermodynamic relation [1]:(1)$$U=N{\left(\frac{V}{N}-b\right)}^{-\frac{1}{c}}\exp\left[\frac{S-Ns_{0}}{cNk_{B}}\right]-a\frac{N^{2}}{V},$$
It binds together thermodynamic state functions, i.e., energy *U*, entropy *S*, volume *V*, and the number of particles *N*, with two interaction parameters *a* and *b*. The number of degrees of freedom of a single molecule is given by constant *c* (c=3/2 for single atoms); $s_{0}$ is a constant that does not appear in the equations of state given further on; $k_{B}$ is the Boltzmann constant.

In equilibrium thermodynamics, *a* and *b* are also parameters of state just like *S*, *V*, and *N* [24,25,26]. Therefore, for the van der Waals gas, they are present in the differential of energy (first law of thermodynamics):(2)$$dU=TdS-pdV-\frac{N^{2}}{V}da+Nk_{B}T{\left(\frac{V}{N}-b\right)}^{-1}db$$
with temperature $T=\partial U\left(S,V,a,b\right)/\partial S$, pressure $p=-\partial U\left(S,V,a,b\right)/\partial V$, $\frac{N^{2}}{V}=-\partial U(S,V$, a,b)/∂a, and $Nk_{B}T{\left(\frac{V}{N}-b\right)}^{-1}=\partial U\left(S,V,a,b\right)/\partial b$ [1]. Each term in the above expression corresponds to one way the energy enters the van der Waals gas. đQ=TdS is the heat; đW=−pdV is the elementary mechanical work when the volume changes; the last two terms represent the work of external sources required to change the strength of interactions. Modifications of an interaction parameter are used, e.g., in the thermodynamic integration methods [27].

In the following sections, we will benefit from the equivalence between the fundamental thermodynamic relation for the van der Waals fluid (Equation 1) and the energy differential (Equation 2) supplemented with the equations of state:
(3a)$$\begin{array}{cc} p & =\frac{nk_{B}T}{1-nb}-an^{2}, \end{array}$$
(3b)$$\begin{array}{cc} u & =cnk_{B}T-an^{2}, \end{array}$$
where n=N/V is the particle density and u=U/V is the energy density. We will see that the van der Waals gas with constant parameters *a* and *b* out of equilibrium is equivalent to the van der Waals equations with effective interaction parameters $a^{*}$ and $b^{*}$, which may vary with the change of the steady heat flow. Further on, we keep *a* and *b* constant.

## 3. Van der Waals Gas in a Heat Flow

We discuss a simplified van der Waals gas (b=0) first. Consider the system schematically shown in Figure 1, a rectangular cavity with a constant amount of particles *N*.

We distinguish two parallel walls separated by a distance *L* in the *z* direction. The walls are kept at temperatures of $T_{1}$ and $T_{2}$. In other directions, we assumed the translational invariance, which constitutes a 1D problem. We assumed the local equilibrium, that is the dynamics of the gas density $n\left(z\right)$ is governed by thermo-hydrodynamic equations: mass continuity, momentum balance, and energy balance equations [5], which are supplemented with the equations of state ([Disp-formula FD3a-entropy-25-01295]):
(4a)$$\begin{array}{cc} p\left(z\right) & =n\left(z\right)k_{B}T\left(z\right)-an{\left(z\right)}^{2}, \end{array}$$
(4b)$$\begin{array}{cc} u\left(z\right) & =cn\left(z\right)k_{B}T\left(z\right)-an{\left(z\right)}^{2} \end{array}$$
valid for every coordinate *z*. In the steady-state, inside the finite 1D segment, the velocity field has to be equal to 0 everywhere. The constant pressure solution $p\left(z\right)=\mathrm{const}$ follows. Another simplification resulting from the stationary condition is the Laplace equation for the temperature profile with the linear solution [28]:(5)$$T\left(z\right)=T_{1}+\left(T_{2}-T_{1}\right)\frac{z}{L}.$$

To determine the concentration profile, we observe that Equation ([Disp-formula FD4a-entropy-25-01295]) written locally, $p=nk_{B}T-an{}^{2}$, is quadratic in density. Thermodynamic stability conditions [1] require that ${\left(\partial p/\partial n\right)}_{T}\geq 0$, which gives $k_{B}T-2an\geq 0$. Therefore, the only physical solution for the density that satisfies ([Disp-formula FD4a-entropy-25-01295]) is given by
(6)$$n\left(z\right)=\frac{k_{B}T\left(z\right)-\sqrt{{\left(k_{B}T\left(z\right)\right)}^{2}-4ap}}{2a},$$
and the stability condition, $k_{B}T\left(z\right)-2an\left(z\right)\geq 0$, with the use of the above expression for $n\left(z\right)$ is reduced to ${\left(k_{B}T\left(z\right)\right)}^{2}\geq 4ap$. Because the pressure in the system is constant and the temperature profile is known, Equations (Equation 5) and (Equation 6) allow us to determine the total number of particles in the system:(7)$$\begin{array}{cc} \; & N\left(T_{1},T_{2},A,L,p\right)=A\int_{0}^{L}dz\;n\left(z\right)=\frac{ALk_{B}\left(T_{1}+T_{2}\right)}{2a}\times \\ \; & \times \left[\frac{1}{2}+\frac{4ap}{k_{B}^{2}\left(T_{2}^{2}-T_{1}^{2}\right)}\int_{k_{B}T_{1}/\sqrt{4ap}}^{k_{B}T_{2}/\sqrt{4ap}}du\;\sqrt{u^{2}-1}\right], \end{array}$$
where *A* is the surface area of the system in the direction of translational invariance. Similarly, from Equation ([Disp-formula FD4b-entropy-25-01295]), we determine the total internal energy:(8)$$\begin{array}{c} U\left(T_{1},T_{2},A,L,p\right)=A\int_{0}^{L}dz\;u\left(z\right)\\ =ALp\left[1+\frac{\left(c-1\right)\sqrt{4ap}}{k_{B}\left(T_{2}-T_{1}\right)}\left(g\left(\frac{k_{B}T_{2}}{\sqrt{4ap}}\right)-g\left(\frac{k_{B}T_{1}}{\sqrt{4ap}}\right)\right)\right] \end{array}$$
with $g\left(x\right)=\frac{1}{3}\left[x^{3}-{\left(x^{2}-1\right)}^{\frac{3}{2}}-1\right]$.

## 4. Net Heat for van der Waals Gas and New Parameter of State

In a steady-state, the same amount of heat enters through one wall and leaves through the other. However, during the transition from one steady-state to another, e.g., by a slight change of temperature $T_{2}$ or by the motion of the right wall changing *L* (see Figure 1), this balance is, in general, disturbed, and the net heat may flow to the system, changing its internal energy [23]. In the case of a very slow transition between stationary states, the energy changes only by means of mechanical work and heat flow: (9)dU=đQ+đW.
We keep the interaction parameters *a* and *b* constant; therefore, there is no term with da and db differentials. The effect related to the change of *a* and *b* is considered in Equation (Equation 2) in equilibrium, but for the steady-state situations, we confined ourselves to the change of the parameters $T_{1},T_{2},A,L$. The mechanical work is given by
(10)đW=−pdV.
and the energy balance during the transition between nonequilibrium steady-states has the following form: (11)dU=đQ−pdV.
The above equation reduces to the first law of thermodynamics in equilibrium. It has the same form, but here, đQ is the net heat transferred to the system during a small change between two stationary instead of equilibrium states.

We obtain the formal analogy between the equilibrium and stationary state for the van der Waals gas by integrating the equations of state ([Disp-formula FD4a-entropy-25-01295]) over the volume:
(12a)$$\begin{array}{cc} pV & =A\int_{0}^{L}dz\;n\left(z\right)k_{B}T\left(z\right)-Aa\int_{0}^{L}dz\;n{\left(z\right)}^{2}, \end{array}$$
(12b)$$\begin{array}{cc} U & =\frac{3}{2}A\int_{0}^{L}dz\;n\left(z\right)k_{B}T\left(z\right)-Aa\int_{0}^{L}dz\;n{\left(z\right)}^{2}, \end{array}$$
and by introducing the average temperature:(13)$$T^{*}\equiv \frac{A\int_{0}^{L}dz\;n\left(z\right)T\left(z\right)}{A\int_{0}^{L}dz\;n\left(z\right)}$$
and the effective potential energy parameter:(14)$$a^{*}\equiv \frac{Aa\int_{0}^{L}dz\;n{\left(z\right)}^{2}}{AL{\bar{n}}^{2}}=\frac{a\int_{0}^{L}dz\;n{\left(z\right)}^{2}}{L{\bar{n}}^{2}},$$
where $\bar{n}=N/V$ is the average particle density and $\bar{u}=U/V$ is the total energy of the system divided by its volume. As a result, we obtain two relations:
(15a)$$\begin{array}{cc} p & =\bar{n}k_{B}T^{*}-a^{*}{\bar{n}}^{2}, \end{array}$$
(15b)$$\begin{array}{cc} \bar{u} & =c\bar{n}k_{B}T^{*}-a^{*}{\bar{n}}^{2}, \end{array}$$
which (for b=0) are formally identical to ([Disp-formula FD3a-entropy-25-01295]). Because Equation ([Disp-formula FD15a-entropy-25-01295]) has the same structure as the equilibrium equation of state, they relate to the fundamental relation (Equation 1):(16)$$U\left(S^{*},V,N,a^{*}\right)=N{\left(\frac{V}{N}\right)}^{-\frac{1}{c}}\exp\left[\frac{S^{*}-Ns_{0}}{cNk_{B}}\right]-a^{*}\frac{N^{2}}{V},$$
but with effective parameters. Moreover, the above equation defines $S^{*}$, and it has a differential:(17)$$dU=T^{*}dS^{*}-pdV-\frac{N^{2}}{V}da^{*},$$
where $T^{*}={\left(\partial U/\partial S^{*}\right)}_{V,N,a^{*}}$, $p={\left(\partial U/\partial V\right)}_{S^{*},N,a^{*}}$ and $\frac{N^{2}}{V}=-\partial U\left(S^{*},V,a^{*}\right)/\partial a^{*}$. Similar to the equilibrium thermodynamics, the steady-state fundamental relation (Equation 16) is not unique, because the modification of $s_{0}$ leads to the same effective equations of state ([Disp-formula FD15a-entropy-25-01295]) and ([Disp-formula FD15b-entropy-25-01295]).

The comparison of Equations (Equation 11) and (Equation 17) gives the relation between the net heat in the system and the effective entropy: (18)$$đQ=T^{*}dS^{*}-\frac{N^{2}}{V}da^{*}.$$
The net heat flow during the transition between two steady-states is a combination of the two exact differentials: effective entropy $dS^{*}$ and effective interaction $da^{*}$. It is contrary to the equilibrium thermodynamics, where the heat is determined solely by the temperature and the change of entropy.

## 5. The Integrating Factor for Net Heat in the van der Waals Gas in Steady-States Does Not Exist

We rearrange Equation (Equation 11) to obtain the net heat: (19)đQ=dU+pdV.
The energy and pressure can be determined from the stationary solution. Therefore, we are in a position to ask whether the heat differential đQ has an integrating factor in space $T_{1},T_{2},V$. For the ideal gas (a=0), the integrating factor exists [23]. It follows that there exists a function of state that is constant if the steady-state system is “adiabatically insulated” (i.e., the net heat vanishes, đQ=0).

We say that a differential form $đF=f_{1}\left(x_{1},x_{2},x_{3}\right)dx_{1}+f_{2}\left(x_{1},x_{2},x_{3}\right)dx_{2}+f_{3}\left(x_{1},x_{2},x_{3}\right)dx_{3}$ has an integrating factor if there exists a function $\phi \left(x_{1},x_{2},x_{3}\right)$ whose differential is related to đF by
$$d\phi \left(x_{1},x_{2},x_{3}\right)\equiv đF/\mu \left(x_{1},x_{2},x_{3}\right).$$
The function μ is called the integrating factor, and ϕ is called the potential of the form đF. The differential form may be considered in different variables, e.g., given by $y_{i}=y_{i}\left(x_{1},x_{2},x_{3}\right)$ for i=1,2,3. We will write in short form as $Y\left(X\right)$. It is straightforward to check that, when the differential form is transformed into new variables, the integrating factor is given by $\mu \left(X\left(Y\right)\right).$ We can choose the most-convenient set of variables to find the integrating factor of a differential form.

We considered the space of the control parameters, $T_{1},T_{2},A,L,N$. It has been used to represent the number of particles, $N=N\left(T_{1},T_{2},A,L,p\right)$, and the energy in the system, $U=U\left(T_{1},T_{2},A,L,p\right)$, given by Equations (Equation 7) and (Equation 8). To simplify further considerations, let us notice that the surface area, *A*, and the length of the system, *L*, always appear in the above relations as a product, V=AL. We can reduce the space of the control parameters to $T_{1},T_{2},V,N$. Because we confined our considerations to a constant number of particles, *N*, we have three parameters, $T_{1},T_{2},V$. However, the natural variables of the differential form (Equation 19) are *U*, *V*. We will use them in the following considerations, and we took $\tau =T_{2}/T_{1}$ as the third parameter.

Suppose that the net heat has the integrating factor. This means that there exists a potential $\phi \left(U,V,\tau \right)$, the differential of is related to the net heat differential by
$$d\phi \left(U,V,\tau \right)\equiv đQ/\mu \left(U,V,\tau \right).$$
By definition, $d\phi =\frac{\partial \phi }{\partial U}dU+\frac{\partial \phi }{\partial V}dV+\frac{\partial \phi }{\partial \tau }d\tau$. On the other hand, the above relation with Equation (Equation 19) gives $d\phi =1/\mu \left(U,V,\tau \right)dU+p\left(U,V,\tau \right)/\mu \left(U,V,\tau \right)dV.$ The equality of the second derivatives for all three independent variables U,V,τ is a necessary condition for the existence of ϕ. It is easy to check that this condition is satisfied only if $p\left(U,V,\tau \right)$ does not depend on τ:$${\left(\frac{\partial p}{\partial \tau }\right)}_{U,V}=0.$$
Equivalently, if ${\left(\partial p/\partial \tau \right)}_{U,V}\neq 0$, then the integrating factor of the net heat does not exist.

The above condition requires the determination of $p\left(U,V,\tau \right)$. The pressure can be determined from Equations (Equation 7) and (Equation 8), which have the following form: $N=N\left(T_{1},T_{2},V,p\right)$ and $U=U\left(T_{1},T_{2},V,p\right)$. The inversion of the former relation would lead to the formula $p=p\left(T_{1},T_{2},V,N\right)$, but we are not able to obtain explicit expression for *p* in terms of elementary functions. However, what we need is not the function itself, but its derivative over τ. Even if a function is given implicitly, its derivative can be explicitly determined with the use of the simple properties of derivatives [1]. We have a similar situation here: although $p\left(U,V,\tau ,N\right)$ with $\tau =T_{2}/T_{1}$ cannot be explicitly determined from $N=N\left(T_{1},T_{2},V,p\right)$ and $U=U\left(T_{1},T_{2},V,p\right)$, its derivative, ${\left(\partial p/\partial \tau \right)}_{U,V}\neq 0$, can be determined explicitly. By using the properties of the derivatives of functions $U=U\left(T_{1},T_{2},V,p\right)$ and $N=N\left(T_{1},T_{2},V,p\right)$, one shows the following property. The derivative ${\left(\partial p/\partial \tau \right)}_{U,V}\neq 0$ does not vanishes, if the following conditions are satisfied:(20)$${\left\{U,N\right\}}_{T_{1},T_{2}}\neq 0$$
and
$$\frac{T_{2}}{T_{1}}{\left\{U,N\right\}}_{p,T_{2}}+{\left\{U,N\right\}}_{p,T_{1}}\neq 0.$$
In the above expressions, the Poisson bracket is defined by ${\left\{f,g\right\}}_{x,y}\equiv \partial f/\partial x\;\partial g/\partial y-\partial g/\partial x\;\partial f/\partial y$. The proof of the above property requires the standard properties of the derivatives under the change of the variables [1] and is omitted here.

It can be directly checked whether the Poisson bracket (Equation 20) does not vanish for functions $U=U\left(T_{1},T_{2},V,p\right)$ and $N=N\left(T_{1},T_{2},V,p\right)$ given by Equations (Equation 7) and (Equation 8). The calculations are straightforward, but cumbersome. To convince the reader that the Poisson bracket (Equation 20) does not vanish, we considered the limit $T_{2}\to T_{1}.$ It gives the following expression:(21)$$\begin{array}{c} \lim_{T_{2}\to T_{1}}\frac{\partial }{\partial T_{2}}{\left\{U,N\right\}}_{T_{1},T_{2}}=\\ =\frac{\left(c-1\right)k_{B}^{3}V^{2}\left(\frac{k_{B}T_{1}}{\sqrt{ap}}-\sqrt{\frac{{\left(k_{B}T_{1}\right)}^{2}}{ap}-4}\right)}{8a^{2}{\left(\frac{{\left(k_{B}T_{1}\right)}^{2}}{ap}-4\right)}^{3/2}}. \end{array}$$
It follows that, even in the neighborhood of the equilibrium state, $T_{2}\approx T_{1}$, the above Poisson bracket does not vanish. As a consequence, the heat differential for the van der Waals gas has no integrating factor. Thus, a function that plays the role of entropy does not exist for the van der Waals gas in a steady-state with heat flow. The representation $đQ=T^{*}dS^{*}$ is impossible.

It is worth emphasizing that, in our previous paper for ideal gas [23], the fundamental relation was introduced due to the integrating factor of heat. We do not see how that method can be generalized to the present case because, as we show in the current paper, such an integrating factor does not exist for the van der Waals gas. One may wonder whether the fundamental relation with the parameters of state can be introduced for the van der Waals gas in a heat flow. However, here, we use a different approach and introduce the fundamental relation by integrating the local equations of state (mapping).

## 6. Global Steady Thermodynamics for van der Waals Gas with b≠0

So far, we have introduced the global steady thermodynamic description for the van der Waals gas given by Equation (Equation 1) with the reduced parameter, b=0. Here, we consider the b≠0 case in which the following equations of state:(22)$$p=\frac{n\left(z\right)k_{B}T\left(z\right)}{1-bn\left(z\right)}-an{\left(z\right)}^{2},$$
(23)$$u\left(z\right)=cn\left(z\right)k_{B}T\left(z\right)-an{\left(z\right)}^{2},$$
describe the van der Waals gas in a stationary state. As before, the pressure in the system is constant. Integration of the above equations over the volume leads to the following relations:(24)$$p=\frac{\bar{n}k_{B}T^{*}}{1-\bar{n}b^{*}}-a^{*}{\bar{n}}^{2},$$
(25)$$\bar{u}=c\bar{n}k_{B}T^{*}-a^{*}{\bar{n}}^{2},$$
where $T^{*}$ and $a^{*}$ are defined by Equations (Equation 13) and (Equation 14) while $b^{*}$ is defined by the following formula:(26)$$\frac{\bar{n}k_{B}T^{*}}{1-\bar{n}b^{*}}=\frac{1}{L}\int_{0}^{L}dz\frac{n\left(z\right)k_{B}T\left(z\right)}{1-bn\left(z\right)}.$$

Equations (Equation 24) and (Equation 25) show that the nonhomogeneous van der Waals gas in a stationary state with a heat flow can be mapped on to the homogeneous van der Waals gas with effective temperature and interaction parameters, $T^{*},a^{*},b^{*}$. Therefore, it has the following fundamental relation:(27)$$U=N{\left(\frac{V}{N}-b^{*}\right)}^{-\frac{1}{c}}\exp\left[\frac{S^{*}-Ns_{0}}{cNk_{B}}\right]-a^{*}\frac{N^{2}}{V},$$
with partial derivatives, $T^{*}=\partial U\left(S^{*},V,a^{*},b^{*}\right)/\partial S^{*}$ and $p=-\partial U\left(S^{*},V,a^{*},b^{*}\right)/\partial V$. The differential of the above fundamental equation gives
(28)$$dU=T^{*}dS^{*}-pdV-\frac{N^{2}}{V}da^{*}+Nk_{B}T^{*}{\left(\frac{V}{N}-b^{*}\right)}^{-1}db^{*}.$$
Using also the expression for the net heat (Equation 19), we identify the heat differential: (29)$$\;\;đQ=T^{*}dS^{*}-\frac{N^{2}}{V}da^{*}+Nk_{B}T^{*}{\left(\frac{V}{N}-b^{*}\right)}^{-1}db^{*}.$$
The above equations describe the energy balance for the van der Waals gas with a heat flow, and they correspond to the first law in equilibrium thermodynamics when the heat flow vanishes.

The parameters $T^{*},a^{*},b^{*}$ defined by Equations (Equation 13), (Equation 14) and (Equation 26) are not independent. To explain it, we keep in mind that, for a given number of particles, three control parameters $T_{1},T_{2},V$ are sufficient to determine the system’s energy, work, and net heat differential. On the other hand, the energy differential in Equation (Equation 28) is given by four parameters, $S^{*},V,a^{*},b^{*}$. It follows that $S^{*},V,a^{*},b^{*}$ are dependent. Consequently, one of these parameters should be determined by the others, e.g., $b^{*}=b^{*}\left(S^{*},V,a^{*}\right)$.

In the above considerations, the van der Waals gas was enclosed between two parallel walls. Control parameters $T_{1}$, $T_{2}$, *V*, and *N* determine the steady-state. In a more-practical situation, the system does not need to be rectangular, and several temperature parameters, $T_{1},\ldots ,T_{k}$, determine the boundary conditions. The same procedure determines the fundamental relation (Equation 27) because it applies to any density and temperature profile. Even in a situation with an arbitrary number of control parameters (k>2), the five parameters of state $S^{*}$, *V*, *N*, $a^{*}$, and $b^{*}$ are sufficient to determine the energy exchange in the system.

Measurements of the temperature and density profiles can determine the effective interaction parameters and temperature $T^{*}$ by Relations (Equation 13), (Equation 14), and (Equation 26). With the measurements of the net heat and Relation (Equation 29), they lead to the determination of the effective entropy change during transitions between steady-states.

Our considerations were focused on the gas phase of the van der Waals fluid. On the other hand, it is known that the van der Waals equation of state describes the phase transition between gas and liquid. The system may have discontinuous density profiles on the gas–liquid interface. The possible appearance of the liquid phase does not affect the derivation of the nonequilibrium fundamental relation. Coexisting phases complicate the solution of thermo-hydrodynamic equations, but do not influence the methodology and the existence of the effective parameters of state.

## 7. Equations of State up to the Second Order in Average Density

We found it impossible to explicitly determine pressure as a function of $T_{1},T_{2},V,N$ from Equation (Equation 7). To gain insight into the general ideas discussed in this paper, it is convenient to deal with analytical expressions. For this reason, we discuss the density expansion of the van der Waals gas equations of state up to the second order. To facilitate this analysis, we used $T_{2}$ and the temperature ratio:(30)$$\rho =\frac{T_{1}}{T_{2}},$$
instead of two temperatures, $T_{1},$$T_{2}$.

We determined the density expansion of the effective parameters of state $T^{*},a^{*},b^{*}$ defined by Equations (Equation 13), (Equation 14) and (Equation 26) from Equations (Equation 5)–(Equation 7). We obtain
(31)$$\begin{array}{cc} T^{*}\left(\rho ,T_{2},\bar{n}\right) & \approx T_{2}\frac{\rho -1}{\log\rho }\left[b\bar{n}\left(\frac{{\left(1-\rho \right)}^{2}}{\rho \log^{2}\rho }-1\right)+1\right]\\ \; & -\frac{a\bar{n}}{k_{B}}\frac{{\left(\rho -1\right)}^{2}\left(\rho^{2}-2\rho \log\rho -1\right)}{2\rho^{2}\log^{3}\rho }+O\left({\bar{n}}^{2}\right), \end{array}$$
(32)$$a^{*}\left(\rho ,T_{2},\bar{n}\right)=a\frac{{\left(1-\rho \right)}^{2}}{\rho \log^{2}\left(\rho \right)}+O\left(\bar{n}\right),$$
(33)$$b^{*}\left(\rho ,T_{2},\bar{n}\right)=b+O\left(\bar{n}\right).$$
Equations (Equation 31)–(Equation 33) are sufficient for the straightforward determination of the van der Waals equations of state up to the second order in density.

## 8. Example of Maxwell Relations

One of the predictive features of equilibrium thermodynamics follows from the equilibrium fundamental relation. The first derivatives of energy, $U\left(S,V,N\right)$, are measurable quantities (cf. pressure as an example), and for this reason, just the symmetry of the second derivatives leads to physical (Maxwell) relations. From these relations, it follows that, for example, for a constant number of particles and interaction parameters, one shows that
(34)$$C_{p}=C_{V}+\frac{TV\alpha^{2}}{\kappa_{T}},$$
which means that heat capacities in constant volume and pressure, $C_{V}\equiv {\left.đQ/dT\right|}_{V,N,a,b}$ and $C_{p}\equiv {\left.đQ/dT\right|}_{p,N,a,b}$, coefficient of thermal expansion, $\alpha =\frac{1}{V}{\left.\partial V/\partial T\right|}_{p,N,a,b}$, and isothermal compressibility, $\kappa_{T}=-\frac{1}{V}{\left.\partial V/\partial p\right|}_{T,N,a,b}$, are not independent [1]. As a result, heat capacity can be determined by the measurements of other quantities, such as isothermal compressibility.

In equilibrium thermodynamics for a constant number of particles, the system’s state is determined by its temperature and density. In a steady-state, the space of control parameters is wider. In particular, for a gas in a box from Figure 1, the space has one dimension higher, $T_{2}$, ρ, $\bar{n}$. It widens the possibilities of thermodynamic processes. For example, for a constant volume and the number of particles in equilibrium, we can change only the temperature of the system. In a steady-state, however, we can change either $T_{1}$ or $T_{2}$ or go along any direction in this two-dimensional space. To demonstrate the predictive possibilities of the steady-state fundamental relation, let us consider a process with a constant temperature ratio ρ for the van der Waals gas in the limit of intermediate densities (up to second order in density expansion). In this situation, Equations (Equation 31)–(Equation 33) determine the effective parameters of state, $T^{*},$$a^{*}$, and $b^{*}$ for given control parameters $T_{1}$, $T_{2}$, $\bar{n}$ or, equivalently, $\rho ,T_{2},\bar{n}$.

If ρ=const, from (Equation 32), it follows that $a^{*}=const$ as well. In this situation, one can define the corresponding measurable quantities that are generalizations of the equilibrium ones as follows: steady-state heat capacities for constant pressure and volume: $$C_{p}^{*}\equiv {\left.\frac{đQ}{dT^{*}}\right|}_{p,N,a^{*},b^{*}},$$
and
$$C_{V}^{*}\equiv {\left.\frac{đQ}{dT^{*}}\right|}_{V,N,a^{*},b^{*}};$$
steady-state coefficient of “thermal” expansion:$$\alpha^{*}=\frac{1}{V}{\left.\frac{\partial V}{\partial T^{*}}\right|}_{p,N,a^{*},b^{*}};$$
steady-state “isothermal” (for constant $T^{*}$) compressibility:$$\kappa_{T^{*}}^{*}=-\frac{1}{V}{\left.\frac{\partial V}{\partial p}\right|}_{T^{*},N,a^{*},b^{*}}.$$
Because the steady-state fundamental relation is the same as in equilibrium, for the thermodynamic path $a^{*}=const$, N=const, we can apply in a straightforward way the reasoning as in equilibrium [1]. This generalizes Equation (Equation 34) to the following expression:(35)$$C_{p}^{*}=C_{V}^{*}+\frac{T^{*}V\alpha^{*2}}{\kappa_{T^{*}}^{*}}.$$
The above formula predicts the relationship between the steady-state properties. In particular, for the case of the van der Waals gas up to second order in density discussed above, both heats should be measured when the temperature ratio ρ is constant, while $T_{2}$ may change. In these conditions, determining $C_{p}^{*}$ requires measuring the excess heat during a small change of $T_{2}$. It is worth mentioning that experimental techniques for the measurements of excess heat have been developed recently [29]. The steady-state “thermal” expansion coefficient should be measured during the change of $T_{2}$ with a constant temperature ratio, ρ=const. The steady-state “isothermal” compressibility should be determined during the pressure change, keeping both temperatures constant.

It is worth noting that the van der Waals equation of state also describes the equilibrium behavior of the gas of interacting particles to the second order in the density expansion. For this reason, the above theory can be tested in molecular dynamics simulations, which give access to the energy, work, heat flux, density, and temperature profiles for a given steady-state and during the transition between nearby steady-states. Moreover, the theory does not rely on the Fourier law for the heat conductivity assumed in this paper, so the assumption about the linearity of the heat flux with the temperature gradient does not need to be controlled in the simulations. Instead, the local equations of state ([Disp-formula FD4a-entropy-25-01295]) and ([Disp-formula FD4b-entropy-25-01295]) must hold in the simulations.

## 9. Discussion

A fundamental relation such as Equation (Equation 1) plays a key role in equilibrium thermodynamics. The fundamental relation, by definition, is a relation between the parameters of the system’s state, from which one can ascertain all relevant thermodynamic information about the system [1]. It includes the identification of different forms of energy exchange of the system with the environment. In equilibrium thermodynamics, the particular terms of the energy differential correspond to heat, mechanical work, or chemical work. In the same spirit, Equation (Equation 27) is the fundamental relation for the van der Waals gas in a steady-state with a heat flow. Its differential (Equation 28) gives information about the net heat and the work performed on the system. Equation (Equation 28) directly reduces to the first law of thermodynamics when the heat flow vanishes. It represents the first law of the global steady thermodynamic description of an interacting system subjected to heat flow.

The integrating factor for the heat differential in the case of the ideal gas discussed previously [23] allowed us to introduce the nonequilibrium entropy and use it to construct the minimum energy principle beyond equilibrium. This principle generalizes thermodynamics’ second law beyond equilibrium. Here, we showed that the net heat has no integrating factor. It excludes a direct generalization of the second law along the line proposed in [23]. However, it does not exclude the possibility that such a principle also exists in the case of an interacting gas.

This paper suggested a general prescription for formulating the fundamental relation of global nonequilibrium steady thermodynamics. First, we identified equilibrium equations of state. Next, we wrote the local equations of state. Whether these equations are in the same form in equilibrium thermodynamics or some other form remains to be found. Next, we averaged these local (or non-local) equations of the state over the entire system. We insist that the global equations of a nonequilibrium state should have the same form as at equilibrium, but with new state parameters. These parameters emerge after averaging the local equations over the entire system. In the case of van der Waals, new state parameters emerged, $a^{*}$ and $b^{*}$. These parameters are constant at equilibrium since they are material parameters that define interactions in a particular system. This result suggests that, in general, all material parameters in the equilibrium equations of states will become parameters of state in the nonequilibrium systems.

The out-of-equilibrium fundamental relation introduced in this paper is formally the same as in equilibrium thermodynamics. It makes it possible to apply the machinery of equilibrium thermodynamics, which “reduces drastically the number of measurements” according to Muller’s quote in the introductory section. The generalization of the thermodynamic Maxwell relations has made it possible to go further along a similar route to equilibrium thermodynamics, this time for an out-of-equilibrium steady-state interacting system.

In equilibrium thermodynamics, every way the system exchanges its energy with the environment is associated with a single parameter of state that appears in the fundamental relation (e.g., heat—entropy, mechanical work—volume, chemical work—number of particles). The example of the van der Waals gas in heat flow shows that the net heat is associated with two or more parameters of state. Fundamental relations for other out-of-equilibrium steady-state systems may, thus, have several parameters corresponding to one way of energy exchange. It suggests further direction of the development. However, a general and systematic method of identifying the nonequilibrium parameters of state and determining the fundamental relation is an open question. 

## Figures and Tables

**Figure 1 entropy-25-01295-f001:**
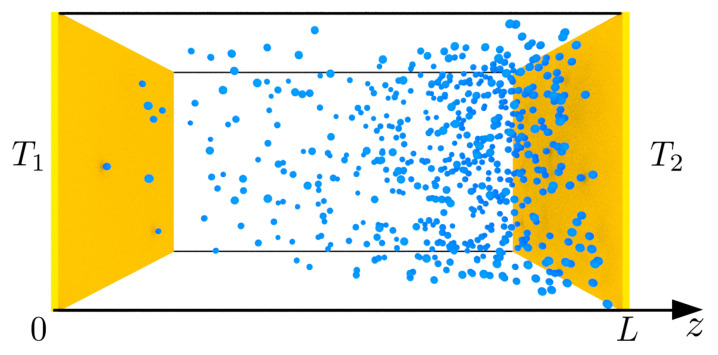
The schematic of the van der Waals gas between parallel walls separated by a distance *L*. The walls are kept at temperatures of $T_{1}>T_{2}$, and the density of the spheres represents the variation of the gas density in the temperature gradient.

## Data Availability

Data is contained within the article.

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
