# Peer review of "Fundamental Relation for Gas of Interacting Particles in a Heat Flow"

_entropy, 2023, doi:10.3390/e25091295_

Round 1

Reviewer 1 Report

The present paper is devoted to deriving a relation that resembles the one obtained in equilibrium thermodynamics for the steady heat flow described by the van der Waals system. By utilizing the previous results on the steady flow in an ideal gas system [23], the authors succeeded in deriving the so-called fundamental relation with only a few parameters. 

The subject of obtaining universal relations to hold in a non-equilibrium steady state is important in the context of steady-state thermodynamics. The present paper is written in a constructive way, and the obtained result is interesting. The reviewer notices only the following points to be resolved: 

1) As the title shows, the authors consider the heat flow in the van der Waals system as a representative gas of interacting particles. However, the van der Waals system is known to describe the phase transition between gas and liquid. Moreover, the system is not always stable depending on the pressure and the temperature, even in equilibrium. Do all relations derived in this paper hold independently of the pressure and the temperature? This point should be discussed and clarified. 

2) In the present paper, the authors derived relations that hold in a specific system and for a specific heat-conduction phenomenon. Of course, the authors may proceed with the same strategy in a different system and/or different phenomena, such as shear flow. However, as thermodynamics is phenomenological, generic relations that hold in any phenomena are highly expected in the future. Please comment on the future outlook of the general construction of the non-equilibrium steady-state thermodynamics. 

In conclusion, the reviewer considers the present paper may be published in Entropy after a suitable revision to resolve the above points 1) and 2).  

Author Response

We are very grateful to Reviewer 1 for the careful reading of the paper. We are glad that the Reviewer found it interesting and written in a constructive way. Below we address all Reviewer’s comments.

Reviewer #1 (Comments to the Author):

The present paper is devoted to deriving a relation that resembles the one obtained in equilibrium thermodynamics for the steady heat flow described by the van der Waals system. By utilizing the previous results on the steady flow in an ideal gas system [23], the authors succeeded in deriving the so-called fundamental relation with only a few parameters.

The subject of obtaining universal relations to hold in a non-equilibrium steady state is important in the context of steady-state thermodynamics. The present paper is written in a constructive way, and the obtained result is interesting. The reviewer notices only the following points to be resolved:

1) As the title shows, the authors consider the heat flow in the van der Waals system as a representative gas of interacting particles. However, the van der Waals system is known to describe the phase transition between gas and liquid. Moreover, the system is not always stable depending on the pressure and the temperature, even in equilibrium. Do all relations derived in this paper hold independently of the pressure and the temperature? This point should be discussed and clarified.

Reply:

To clarify the above point we added the following paragraph at the end of section 6:

Our considerations are focused on the gas phase of van der Waals fluid. On the other hand, it is known that the van der Waals equation of state describes the phase transition between gas and liquid. The system may have discontinuous density profiles on the gas-liquid interface. A possible appearance of the liquid phase does not affect the derivation of the nonequilibrium fundamental relation. Coexisting phases complicate the solution of thermo-hydrodynamic equations but do not influence the methodology and the existence of the effective parameters of state.

Reviewer #1 (Comments to the Author):

2) In the present paper, the authors derived relations that hold in a specific system and for a specific heat-conduction phenomenon. Of course, the authors may proceed with the same strategy in a different system and/or different phenomena, such as shear flow. However, as thermodynamics is phenomenological, generic relations that hold in any phenomena are highly expected in the future. Please comment on the future outlook of the general construction of the non-equilibrium steady-state thermodynamics.

Reply:

At the end of our manuscript we added the following comment:

In equilibrium thermodynamics, every way the system exchanges its energy with the environment is associated with a single parameter of state that appears in the fundamental relation (e.g., heat – entropy, mechanical work – volume, chemical work – number of particles). The example of van der Waals gas in heat flow shows that the net heat is associated with two or more parameters of state. Fundamental relations for other out-of-equilibrium steady state systems may thus have several parameters corresponding to one way of energy exchange. It suggests further direction of the development. However, a general and systematic method of identifying the nonequilibrium parameters of state and determining the fundamental relation is an open question.

Reviewer 2 Report

SUMMARY:

This manuscript is an extension of previous work by the authors in which a description of non-equilibrium systems in steady states in terms of global thermodynamics functions is proposed. As an extension of the results presented in that previous work, which concentrated on the ideal gas in a heat flow, this manuscript explores whether such a description exists for a van der Waals gas in a heat flow. The new results present a fundamental relation for the later situation of interacting particles. It is also shown that the net heat flow has not an integrating factor, so non-equilibrium entropy can not be introduced in the same manner as in the previous work for ideal gases. The manuscript nevertheless presents an approach to formulate such fundamental relation for the van der Waals gas under heat flow. Thermodynamics relations derived from this formalism are discussed, along with possible experimental verifications.

COMMENTS:

The manuscript reads well, and the calculations are convincing. It is indeed a very interesting extension of previously proposed formalism by the authors. I notice that some comments about possible experimental verification of the predictions of the formalism are added; however, it could also be interesting to add some comments about possible computational verification, like using molecular dynamics. Some comments about the validity of the local equilibrium assumption should also be added.

RECOMMENDATIONS:

I think this manuscript can be published after the authors address my comments. The manuscript is clear, and as mentioned above, it reads well, and the reported results could be of interest to the readers of this journal.

The quality of English language is good.

Author Response

We thank the Reviewer 2 for examining our manuscript. We are glad that the Reviewer found it well-written and the calculations convincing. Below we answer the Reviewer’s comments.

Reviewer #2 (SUMMARY):

This manuscript is an extension of previous work by the authors in which a description of non-equilibrium systems in steady states in terms of global thermodynamics functions is proposed. As an extension of the results presented in that previous work, which concentrated on the ideal gas in a heat flow, this manuscript explores whether such a description exists for a van der Waals gas in a heat flow. The new results present a fundamental relation for the later situation of interacting particles. It is also shown that the net heat flow has not an integrating factor, so non-equilibrium entropy can not be introduced in the same manner as in the previous work for ideal gases. The manuscript nevertheless presents an approach to formulate such fundamental relation for the van der Waals gas under heat flow. Thermodynamics relations derived from this formalism are discussed, along with possible experimental verifications.

Reviewer #2 (Comments to the Authors):

The manuscript reads well, and the calculations are convincing. It is indeed a very interesting extension of previously proposed formalism by the authors. I notice that some comments about possible experimental verification of the predictions of the formalism are added; however, it could also be interesting to add some comments about possible computational verification, like using molecular dynamics. Some comments about the validity of the local equilibrium assumption should also be added.

Reply:

To clarify the above two comments we added at the end of section 8:

It is worth noting that the van der Waals equation of state also describes the equilibrium behavior of the gas of interacting particles to the second order in the density expansion. For this reason, the above theory can be tested in molecular dynamics simulations, which give access to the energy, work, heat flux, density, and temperature profiles for a given steady state and during the transition between nearby steady states. Moreover, the theory does not rely on the Fourier law for the heat conductivity assumed in this paper, so the assumption about the linearity of the heat flux with the temperature gradient does not need to be controlled in the simulations. Instead, the local equations of state (4a) and (4b) must hold in the simulations.